# Patients with Colorectal Cancer and *BRAF^V600E^*-Mutation in Argentina: A Real-World Study—The EMOGI-CRC01 Study

**DOI:** 10.3390/cancers17061007

**Published:** 2025-03-17

**Authors:** Greta Catani, Stefano Kim, Federico Waisberg, Diego Enrico, Romina Luca, Federico Esteso, Luisina Bruno, Andrés Rodríguez, Marcos Bortz, Berenice Freile, Matías Chacón, Ana Isabel Oviedo Albor, Guillermo Méndez, Ezequiel Slutsky, María Cristina Baiud, Romina Llanos, Ayelen Solonyezny, Luis Basbus, Gerardo Arroyo, Julieta Grasselli, Rosario Pasquinelli, Luciana Bella Quero, María Victoria Faura, Ana Cecilia Adur, Mariano Dioca, Mercedes Tamburelli, Javier Castillo, Juan Manuel O’Connor

**Affiliations:** 1Department of Oncology, Alexander Fleming Institute, Buenos Aires 1426, Argentina; fwaisberg12@gmail.com (F.W.); diego-enrico@hotmail.com (D.E.); marcosbortz@gmail.com (M.B.); matiemi@yahoo.com (M.C.); 2Equipo Multidisciplinario de Oncología Gastrointestinal (EMOGI), Asociación Argentina de Oncología Clínica, Federico Lacroze 2252, Argentina; stefanokim@me.com (S.K.); federico.esteso@yahoo.com (F.E.); luisina_bruno@hotmail.com (L.B.); andrwrodri84@gmail.com (A.R.); berefreile@gmail.com (B.F.); draanyoviedo1612@gmail.com (A.I.O.A.); mendezdoc@hotmail.com (G.M.); luis.basbus@hospitalitaliano.org.ar (L.B.); ga255538@gmail.com (G.A.); julietagrasselli@gmail.com (J.G.); lubq1@yahoo.com.ar (L.B.Q.); mdioca@gmail.com (M.D.); mtamburelli@hospitalaleman.com (M.T.); javierocastillo@gmail.com (J.C.); juanmanueloconnor@gmail.com (J.M.O.); 3Department of Oncology, Sanatorio Allende, Córdoba X5000BFB, Argentina; cris_baiud@hotmail.com (M.C.B.); llanosrominab@gmail.com (R.L.); solonyeznya@gmail.com (A.S.); 4European Oncology Institute Strasbourg, Department Medical Oncology, CEDEX 67200 Strasbourg, France; 5Department of Gastrointestinal Tumors, Alexander Fleming Institute, Buenos Aires 1426, Argentina; rominaluca23@gmail.com; 6Department of Gastrointestinal Tumors, Dr. C.B. Udaondo Hospital, Buenos Aires 1264, Argentina; 7Department of Oncology, Favaloro Foundation, Buenos Aires 1093, Argentina; eslutsky@gmail.com; 8Department of Oncology, Hospital Italiano Buenos Aires, Buenos Aires 1199, Argentina; 9Department of Oncology, Centro de Diagnóstico, Investigación y Tratamiento, Salta 4400, Argentina; 10Department of Oncology, CEMIC, Buenos Aires 1431, Argentina; pasquinellirosario@gmail.com; 11Department of Oncology, Hospital Británico, Buenos Aires 1280, Argentina; mvictoriafaura@gmail.com; 12Department of Gastrointestinal Tumors, Instituto de Oncología Ángel H. Roffo, Buenos Aires 1417, Argentina; aadur22@gmail.com; 13Department of Oncology, Hospital Alemán, Buenos Aires 1425, Argentina

**Keywords:** metastatic colorectal cancer, *BRAF^V600E^*-mutation, real-world data, Latin America

## Abstract

*BRAF*-mutated colorectal cancer is an aggressive entity related with poor prognoses and limited data in Latin America, reflecting gaps in research and practice. The EMOGI-CRC01 study analyzed 161 patients with *BRAF^V600E^*-mutated metastatic colorectal cancer (mCRC) across ten oncology centers in Argentina. The median age was 58.5 years, and 93.8% of patients received first-line treatment, mostly doublet chemotherapy plus anti-VEGF. With a median follow-up of 23 months, the progression-free survival (PFS) was 9 months. In second-line treatment, only 26% received BRAF inhibitors, with median PFS 5.2 months (95% CI 4.9—NR) and the overall response rate (ORR) was 10.5%. Real-world studies are essential to understand the barriers. Our study highlights treatment heterogeneity due to limited access to high-cost drugs, especially in the second-line setting.

## 1. Introduction

The incidence and mortality of colorectal cancer (CRC) have been steadily rising in many countries, especially in Eastern Europe, Asia, and South America, over the last 20 years [1]. In Argentina, CRC is the second most common malignant tumor in both sexes, with 15,895 new cases in 2020, and is the second leading cause of cancer mortality [2]. Approximately 22% of CRC patients are diagnosed in advanced stages, with a 5-year overall survival (OS) of 15% [3]. In recent years, identifying various tumor genomic mutations in metastatic colorectal cancer (mCRC) has provided information to guide treatment strategies. In this setting, analyses of *KRAS*, *NRAS*, and *BRAF* mutations, as well as HER2 amplification and microsatellite instability (MSI) or mismatch repair protein (MMR) status, are recommended in daily practice by international societies and guidelines [4,5].

Around 8 to 12% of cases with advanced CRC and 14% of those with localized CRC have a BRAF mutation [5,6]. This mutation is classically more frequent in female patients, right-sided tumors, advanced-stage disease, mucinous histology, and MMR pathway alterations. Importantly, these mutations have been associated with a poor prognosis in mCRC, with a median OS of 10 to 16 months [7]. Despite this, *BRAF^V600E^*-mutant mCRC tumors are emerging as distinct biological entities characterized by clinical and molecular heterogeneity [8].

An optimal treatment sequence in this population has not been established. However, current guidelines propose that the preferred first-line treatment strategy should include a doublet or triplet combination chemotherapy regimen, with or without the vascular endothelial growth factor (VEGF) inhibitor bevacizumab [4,5]. In the second-line setting, the doublet combination of a BRAF and EGFR (with or without MEK) inhibitor evaluated in the phase 3 BEACON trial resulted in significantly longer OS and a higher response rate than standard therapy in patients with the *BRAF^V600E^*-mutation [9]. Based on these results, it became the standard of care in this scenario. In Argentina, this combination was approved in April 2023.

Nevertheless, access to high-cost drugs is limited, given that only 60% of our population has private or social security health insurance that can afford these drugs [10]. As a consequence, the timely treatment of patients with *BRAF^V600E^*-mutated tumors is challenging in most cancer health centers.

Considering the need for a deeper understanding of this patient subgroup, and the diverse therapeutic options currently available, we aimed to describe the baseline characteristics, analyze the treatment strategies, and assess the outcomes in patients with CRC with *BRAF^V600E^*-mutation across multiple oncology centers in Argentina. This study provides an opportunity to identify and reduce gaps in current evidence.

## 2. Materials and Methods

### 2.1. Study Design

This was a multicenter, retrospective, observational cohort study that included patients with advanced CRC with *BRAF^V600E^*-mutation treated in ten public and private institutions in Argentina from January 2014 to June 2023. Patients’ information was collected from medical charts and entered into a common anonymized database. *BRAF^V600E^*-mutation status was tested using q-PCR, PCR-real time, or next-generation sequencing (NGS) in tissue or liquid biopsy analysis.

### 2.2. Key Study Endpoints

The primary objective was to evaluate the progression-free survival (PFS) of the first- and second-line treatment for each treatment received in this setting. PFS was defined as the time from the start of first- or second-line treatment to the first evidence of disease progression (or the start of a subsequent line of treatment, as applicable) or death due to any cause (whichever occurs first). Secondary objectives were to describe clinical-pathological characteristics, with particular attention to social and demographic factors of the population, and the evaluation of objective response rate (ORR) according to investigators following the RECIST 1.1 criteria.

### 2.3. Statistical Analyses

All of the data collected from medical records were analyzed using descriptive statistics, including proportions, medians, interquartile ranges, means, and confidence intervals, when appropriate. Survival curves were generated using the Kaplan–Meier method, and differences between groups were calculated using the log-rank test.

Cox proportional hazard models with a stepwise procedure were used to estimate the hazard ratio and corresponding 95% CIs [11]. Univariate analyses were conducted to identify potential prognostic factors for PFS. Subsequently, multivariate analyses for PFS were performed, initially including factors with a significance level of *p* < 0.2 from the univariate model. The final model incorporated only variates associated with a PFS with a significance threshold below 0.05.

The data management and statistical calculations were performed with R software version 4.0.3 and IBM SPSS Version 26.0 Statistics Editor Data.

## 3. Results

### 3.1. Patient’s Characteristics

A total of 161 patients with *BRAF^V600E^*-mutated CRC at the advanced setting were included in the explorative analysis. The median age at diagnosis was 58.5 (IQR 47–69), and 59.6% (n = 96) were female. Right-sided primary localization was the most prevalent with 70.8% (n = 114) and 85.7% (n = 135) having ECOG 0–1 at the time of disease presentation. Of note, 71.4% (n = 115) were diagnosed at an advanced stage. The most common metastatic localizations included the liver with/without another metastatic site in 55.9% (n = 90), followed by the lymph nodes in 50.9% (n = 82), peritoneum in 23% (n = 37), and lungs in 11.8% (n = 19) (Table 1).

In the overall population, 100% (n = 161) of the tumors had *BRAF^V600E^*-mutation. We found that of the 89.5% (n = 144) of the cases tested using tumoral tissue samples, 83.8% (n = 134) were characterized as RAS (KRAS, NRAS) wild-type, and 1.2% (n = 3) RAS-mutated (Table 1). In total, 94.4% of the patients included (n = 152) were tested for MMR status by immunohistochemistry, and 21.7% (n = 35) of them were characterized as MMR-deficient (dMMR) tumors. In this subgroup, 71.9% (n = 23) of cases had an absence of MLH1/PMS-2 protein expression.

Initial surgery was indicated in 47.8% (n = 77). Different types of surgical procedures were performed, such as hemicolectomy, colectomy, and metastasectomy. Palliative resections were performed in 13.6% (n = 22) of the total cases, including metastases or primary tumor resection to reduce tumor burden or to mitigate symptoms. Adjuvant therapy was received in 20.5% (n = 33) of all the population included.

### 3.2. Treatment Sequences

We next investigated treatment sequences and efficacy in the advanced disease setting. Ten patients were unable to start treatment due to poor performance status, requiring hospitalization, or because their oncologist considered them unfit for systemic therapy. Among 93.8% (n = 151) of the patients, access to first-line treatment was possible (Table 1). The most common strategies in this context were doublet chemotherapy [FOLFOX, XELOX, or FOLFIRI (5-fluorouracil plus leucovorin plus irinotecan)] plus anti-VEGF therapy in 49% (n = 74), followed by doublet chemotherapy alone in 19.2% (n = 29) and triplet therapy FOLFOXIRI (5-fluorouracil plus leucovorin plus irinotecan plus oxaliplatin) plus anti-VEGF in 13.2% (n = 20). Furthermore, 72.8% (n = 110) received doublet chemotherapy with or without targeted therapy and only 14.5% (n = 22) received triplet chemotherapy with or without targeted therapy Figure 1. These two subgroups present different clinical characteristics: the median age was 59 (IQR 47–69) versus 52 years (IQR 43–61), 10% (n = 11) versus 18.2% (n = 4) presented three or more metastatic sites, and dMMR tumors were detected in 17.3% (n = 19) versus 4.5% (n = 1), respectively. Other treatment strategies are summarized in Figure 1.

The proportion of patients in the full analysis set receiving each treatment type is shown in Figure 1. The number of patients per treatment regimen is provided within each bar.

### 3.3. Treatment Strategies and Outcomes in First- and Second-Line Therapy

With a median follow-up of 23 months, the median OS was 20.6 months (95% CI 18.4–29 months) and median PFS was 9 months (95% CI 7.4–10.5 months) (Figure 2). Conforming different treatments in these circumstances, the median PFS for chemotherapy plus targeted therapy was 9 months (95% CI 7.1–10.8 months), chemotherapy alone was 10 months (95% CI 6.9–13 months) and for immune checkpoint inhibitors (ICI) was NR (95% CI NR) (*p* = 0.079) (Figure 3a). Median PFS was 4 months (95% CI 2.9–5.1 months) and 7 months (95% CI 5.5–8.4 months) [HR 1.4 95% (CI 0.8–2.3); *p* = 0.16] for patients who received triplet to doublet chemotherapy with or without targeted therapy, respectively (Figure 3b).

After first-line therapy, 75.5% (n = 114) of patients with advanced disease experienced progression at the time of analysis. The main locations included the liver in 53.5% (n = 61), the peritoneum in 46.9% (n = 53), lymph nodes in 18.4% (n = 21), and lungs in 8.7% (n = 10). Among them, 48.6% (n = 86) underwent a subsequent systemic therapy.

Subsequent multivariate analysis found that male gender, peritoneum involvement, and absence of surgery on the primary tumor were significant prognostic factors for PFS (*p* = 0.06, 0.01, and 0.01, respectively) (Table 2). The univariate analysis data are provided in Table 2.

In this context, in the second-line treatment, the median PFS was 5 months (95% CI 4.1–7.07 months). The treatments administered in this setting were heterogeneous. The most common regimens were doublet chemotherapy plus anti-EGFR in 29% (n = 25), BRAFi in 26% (n = 19) and doublet chemotherapy plus anti-EGFR in 11.9% (n = 10) (Figure 1). Median PFS was 4.2 months (95% CI 3.5–8.2 months) for patients who received targeted therapy with or without chemotherapy (n = 46), 2.6 months (95% CI 1.9—NR) for chemotherapy alone (n = 11), NR (95% CI 1.6—NR) for ICI (n = 7) and 5.2 months (95% CI 4.9—NR) for the subgroup who received BRAFi (n = 19) (log-rank *p* = 0.043) (Figure 3c). Subsequent treatments can be found in Figure 1.

### 3.4. Immunotherapy and BRAF Inhibitors

Considering the total number of patients included in the treatment analysis (n = 151), 15.2% (n = 23) received ICI: 43.5% (n = 10) in the first-line, 30.4% (n = 7) in the second-line, and 26.1% (n = 6) in the third-line setting. Among them, 93.1% (n = 21) patients received pembrolizumab, while only 6.9% (n = 2) received nivolumab/ipilimumab. All these patients were dMMR. In both the first- and second-line settings, the PFS was NR (Figure 3a–c).

According to MMR status, the median OS was 20.1 months (95% CI 13.5–25.3 months) for MMR-proficient (pMMR) and 36 months (95% CI 16.0–55.9 months) for dMMR (HR 1.67 [95% CI 0.99–2.8]; *p* = 0.12) with a median FU of 20.1 months (Figure 4). Also, we compared PFS of patients with dMMR tumors who received ICI or other than ICI treatments in the first-line setting. The median PFS was NR (95% CI NR) for ICI and 6 months (95% CI 4.4–7.5 months) for non-ICI (HR 0.38 [95% CI 0.1–1.1]; *p* = 0.062).

BRAFi were mostly used in the second-line setting in 26% (n = 19). Regimens with BRAFi included encorafenib plus cetuximab in 56% (n = 14), vemurafenib plus irinotecan plus cetuximab in 32% (n = 8), encorafenib plus cetuximab plus binimetinib in 4% (n = 1), vemurafenib plus irinotecan in 4% (n = 1) and vemurafenib plus cetuximab 4% (n = 1) of the cases (Figure 1). The ORR for those who received BRAFi was 10.5% and median PFS 5.2 months (95% CI 4.9—NR) in the second-line setting.

## 4. Discussion

*BRAF^V600E^*-mutation mCRC is a highly aggressive and biologically heterogeneous disease, associated with poor prognoses and limited therapeutic options. In this real-world multicenter study, we analyzed the clinical characteristics, treatment strategies, and outcomes of 161 patients with *BRAF^V600E^*-mutation mCRC in Argentina. Our findings highlight significant variability in treatment approaches, with limited access to BRAF-targeted therapies in second- and third-line settings. Despite an mOS of 20.6 months, disease progression remained a major challenge. Notably, while 93.8% of patients received first-line treatment, only 15.5% were able to undergo third-line therapy, reflecting the aggressive nature of the disease and the rapid clinical deterioration that limits the feasibility of subsequent treatment lines.

In our cohort, 28% of the included patients were diagnosed at an early stage (stage I-III) but experienced metastatic progression during the course of the disease. For this reason, several ongoing clinical trials are investigating the use of BRAF-targeted therapies in this setting. These studies aim to determine whether incorporating targeted treatments can improve outcomes in this potentially curable population. The Alliance A022004 trial (NCT05710406), a phase II/III study, evaluated the efficacy of the combination of encorafenib and cetuximab after surgery and chemotherapy. AIO-KRK-0420 (NCT05510895) is a phase II trial investigating the efficacy of a neoadjuvant regimen combining encorafenib, binimetinib, and cetuximab. The NEORAF study (NCT05706779), a phase II study, is evaluating the effectiveness of neoadjuvant treatment with encorafenib plus cetuximab in patients with operable *BRAF^V600E^*-mutant adenocarcinoma of the colon or upper rectum [12,13,14]. The results of these three trials have not been published yet, but they reflect a growing interest in integrating BRAF-targeted therapies into the treatment paradigm for earlier-stage *BRAF*-mutant CRC.

Patients with dMMR-*BRAF*-mutant tumors constitute a very small subgroup, accounting for about 1–2% of mCRC, and the prognostic impact of the MMR status in the subgroup of BRAFV600E mutant mCRC has been poorly studied [15,16]. Venderbosch and colleagues pooled the results of four phase III trials and suggested that the adverse prognosis associated with dMMR is influenced by the patient’s BRAF-mutation status [17]. In our study, patients with dMMR tumors had better OS (compared to pMMR) and better PFS if they received ICI (compared to non-ICI treatment) in the first- and second-line setting. We could explain this because in the analysis by Venderbosch et al., patients with *BRAF*-mutated and dMMR tumors did not receive ICI, which are now recognized as a standard and highly effective treatment for this subgroup. In contrast, in our study, these patients did receive ICI, which could explain their improved prognosis compared to historical cohorts, despite the typically poor outcomes associated with *BRAF*-mutation. Also, in the context of *BRAF*-mutant CRC, the activation of the MAPK signaling pathway, induced by the *BRAF*-mutation, may contribute to increased expression of immune evasion-related proteins, such as PD-L1, thereby enhancing the tumor’s vulnerability to immunotherapy. Thus, the combination of a higher mutational burden in dMMR tumors and alterations in the MAPK signaling pathway facilitates the efficacy of immunotherapeutic agents, which, by interfering with immune checkpoints, allow for a more robust and effective immune response against the tumor [18,19].

The efficacy results showed some differences between our study and literature. In the CAPSTAN CRC study, a real-world study that included 255 patients with *BRAF^V600E^*-mutant mCRC from European countries, the median OS was 12.9 months (95% CI 11.6–14.1) and mPFS 6.0 months (95% CI 5.3–6.7) across all regimens in the first-line setting [20]. Another real-world study conducted by Xu et al. included 261 patients with BRAFV600E-mutant mCRC and showed a median OS of 18.2 months (95% CI 16.4–20) and median PFS of 6.4 months [21].

Notably, the median OS and PFS were higher in our cohort in contrast to findings from previous reports in clinical trials. Several explanations might account for this difference: availability of ICI in the dMMR group, disparities in drug access, and the diverse treatments indicated. Also, our cohort may have included a more favorable patient population, such as those with better baseline performance status or more localized disease at diagnosis, which could have led to better outcomes. Additionally, the size of our cohort, while sufficient for meaningful analysis, may have resulted in a more homogeneous group in terms of treatment adherence and follow-up, which can influence survival outcomes. Furthermore, the majority of the patients included in our cohort were followed up in the private healthcare sector, where patient monitoring tends to be more stringent. Unfortunately, in our country, there is a significant gap between the public and private healthcare systems, particularly regarding the availability of human resources and physical resources, such as medications, hospital infrastructure, medical equipment, and facilities, which can directly impact patient care and survival outcomes. However, a recent study by O’Connor et al. showcased a notable increase in BRAF testing in our country, rising from 18% to 64% [22]. These findings were linked to the inclusion of BRAF testing in the gene panel, supported by the pharmaceutical industry.

In our study, in the first-line setting, the main strategy involved doublet chemotherapy plus anti-VEGF. Similar to our results, the CAPSTAN CRC study found that most patients received doublet chemotherapy (74.5%), either alone (28.2%) or in combination with a targeted therapy (46.3%) in the first-line setting [20]. Triplet chemotherapy was indicated in 18.8% in our cohort, and according to the current evidence, triplet chemotherapy (FOLFOXIRI) plus bevacizumab has been frequently considered as a preferred regimen in this scenario, based on the phase III TRIBE study. In this trial, patients with *BRAF^V600E^*-mutant mCRC who received FOLFOXIRI plus bevacizumab in the first-line treatment obtained better outcomes than patients who received FOLFIRI plus bevacizumab (mOS 19 versus 10.7 months, and mPFS 7.5 versus 5.5 months, respectively) [23]. However, these results are still under debate, since a meta-analysis conducted by Cremolini and collaborators did not find increased benefit from adding irinotecan to a first-line strategy based on FOLFOX and bevacizumab [24]. Our results demonstrate a trend toward better PFS with doublet chemotherapy compared to triplet chemotherapy (7 vs. 4 months, respectively; *p* = 0.16). This could possibly be explained by the significant difference in the number of patients receiving doublet chemotherapy compared to triplet chemotherapy, as well as a higher proportion of dMMR tumors and fewer metastatic sites in the doublet group. Therefore, it is probable that higher-risk patients were selected for triplet chemotherapy due to a greater tumor burden and the presence of more metastatic sites, factors that may have influenced their prognoses.

Another important point to highlight is the prevalence of anti-EGFR indication in the first and second line (6% and 8%, respectively). The presence of *BRAF^V600E^*-mutations has been identified as a negative predictor of response to anti-EGFR therapies in patients with mCRC [25]. In both of the phase III trials, CRYSTAL [26] and FIRE-3 [27] demonstrated no benefit with the addition of cetuximab in these specific subgroup (median PFS 8.0 versus 5.6 months; HR = 0.934; *p* = 0.87 and median PFS 6.6 versus 6.6 months; HR = 0.84, *p* = 0.56), respectively. The fact that only 34.7% of physicians were aware of the patient’s BRAF status before making the first-line treatment decision highlights a significant limitation in therapeutic decision-making. The lack of this crucial information hinders the application of a treatment algorithm based on molecular biology, which could have influenced the selection of more personalized and effective treatments, especially in cases of colorectal cancer with BRAF mutations. Without access to this data, physicians may be limited to standard options, reducing the ability to optimize patient outcomes.

During the last 3 years, the advent of BRAFi has redefined the landscape of *BRAF^V600E^*-mutant mCRC. The phase III BEACON trial evaluated encorafenib plus cetuximab with or without the MEK inhibitor binimetinib versus the investigators’ choice of irinotecan or FOLFIRI plus cetuximab in patients that had received one or two prior systemic therapies in the advanced disease setting [9]. Notably, the combination of encorafenib and cetuximab with and without binimetinib resulted in significantly longer OS (9 and 8.4 months, respectively) than standard therapy (5.4 months). As a consequence, the Food and Drug Administration (FDA) and European Medicines Agency (EMA) approved this regimen in patients with *BRAF^V600E^*-mutant mCRC in May 2020). At the present, patients should be treated with encorafenib plus cetuximab as soon as possible when progression to first-line treatment occurs (in second-line rather than later lines) [28]. Latin America (LATAM) is a diverse region with heterogeneous health systems, with barriers in knowledge, options for treatment, equitable distribution, and timely outcomes [29]. In our country, this treatment was approved in April 2023. Before this date, access to BRAFi was more limited than now, and the indication was off-label. However, the high cost of the treatment continues to limit access to this targeted therapy and is directly dependent on the patient’s health insurance. This may explain the heterogeneity of BRAFi regimens administered.

Over time, various attempts have been made to identify predictive and prognostic biomarkers in this heterogeneous disease. In 2019, Loupakis et al. [30] built two different scoring systems, a ‘complete’ score (0–16) including all significant covariates and a ‘simplified’ score (0–9), based only on clinicopathological covariates and excluding laboratory values. After that, the scores were classified into three categories according to the score (also in ‘complete’ and ‘simplified’): low-, intermediate-, and high-risk. The median OS were 26.2 months (IQR 23.0–30.9) for low-risk, 15.7 months (IQR 14.6–18.7) for intermediate-risk, and 6.1 months (IQR 3.8–7.4) for the high-risk in the ‘complete’ score. Similar results occurred in the ‘simplified’ score [30].

Multiple mechanisms of resistance to BRAFi were studied, and recently other pathways that may be involved have been described, such as WNT/b-catenin, an independent MAPK reactivation [31,32,33]. Elez et al. have found that inactivating mutations on RNF43 (a tumor suppressor gene involved in the Wnt/β-catenin signaling pathway) predict response rates and survival outcomes in patients with BRAF-MSS tumors who were treated with BRAFi and anti-EGFR [34]. A similar mechanism of RNF43 loss-dependent WNT activation may be restraining MAPK signaling in MSS mCRC tumors and synergizing with a pharmacological blockade of the pathway. They found a clinical benefit in PFS (10.1 versus 4.1 months) and in OS (13. 6 versus 7 months) in patients with MSS-RNF43 mutated versus MSS-RNF43 wild-type, respectively. This data suggests that RNF43-mutation might be a predictive biomarker of response in patients with *BRAF^V600E^*-mutant mCRC who receive BRAFi plus anti-EGFR treatment. In 2023, Ros et al. showed that the plasmatic BRAFV600E allele fraction (AF) OS was significantly longer in patients with low BRAF AF (< 2%) than in those with high AF (*p* < 0.0001) Also, patients with high-BRAF AF presented benefit from triplet therapy (HR 0.17, 95% CI 0.06–0.53; interaction test = 0.002) [35]. This suggests BRAF AF as a potential predictive factor for assessing the benefits of triplet therapy.

There are some limitations to this study. The limited number of cases treated in a heterogeneous health system characterized by different access to oncological treatments may be different from other low-middle-income countries. Furthermore, we included patients who were diagnosed and treated between 2014 to 2023. During these nine years, different treatment strategies and drugs were implemented in clinical practice. Patients with initially localized disease are also described in our series, which gives an estimation of this driver in our population, but it should be pointed out that this alteration has not been defined as a prognostic factor in this context. At the moment, a number of clinical trials are being carried out to demonstrate the role of *BRAF*-mutation in the localized setting.

Treatment disparity highlights the need to adequately define specific subgroups that may benefit the most from high-cost treatment recommendations. Based on this, a better understanding of the activation of biological pathways, as well as the emergence of resistance mechanisms, may help to select treatment more appropriately. This study represents the first large LATAM real-world effort analyzing patients with *BRAF^V600E^*-mutant CRC. Our results reflect different treatment strategies in a real-life scenario, frequencies by which therapies such as BRAFi, anti-EGFR, anti-VEGF, and ICI are used in mCRC. Treatment decisions for this patient subpopulation are complex and heterogeneous and may vary according to drug availability, social insurance coverage, and patient preferences and comorbidities. The creation of a national registry program is essential to overcome the barrier of obtaining robust and reliable cancer clinical data of patients with tumors associated with infrequent mutations.

## 5. Conclusions

In this large Latin American cohort, a longer OS and PFS were observed compared to previous studies. Our study highlights the diversity of therapeutic indications in the different lines of treatments, especially in the second-line setting. Real-world studies are essential to understanding the barriers and limitations of our daily clinical practice and supporting the development and implementation of health policies to improve patient outcomes. In this context, initiatives such as the creation of a national registry program become essential to overcome the barrier of obtaining robust and reliable cancer clinical data.

The *BRAF^V600E^*-mutation serves as a crucial biomarker offering valuable information for both prognosis and prediction in mCRC. In the context of contemporary evidence, screening for this mutation in the advanced setting is now strongly recommended, together with the assessment of RAS alterations and MMR/MSI status.

## Figures and Tables

**Figure 1 cancers-17-01007-f001:**
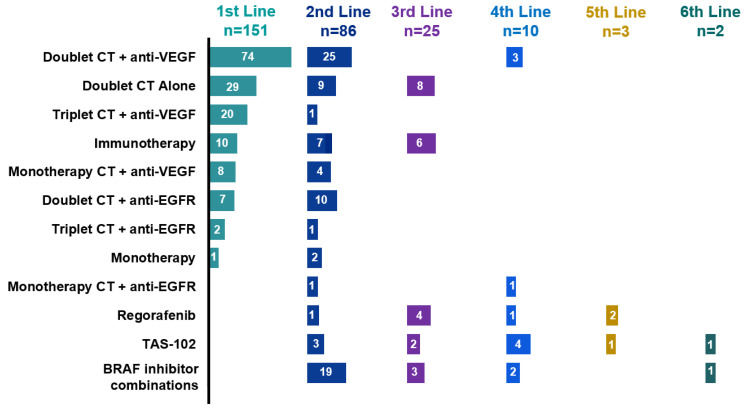
Therapeutics according to treatment line. Doublet CT, doublet chemotherapy (FOLFOX, XELOX, FOLFIRI); triplet CT, triplet chemotherapy (FOLFIRINOX); monotherapy CT, monochemotherapy (fluoropyrimidine, irinotecan); anti-EGFR (epidermal growth factor receptor), cetuximab or panitumumab; anti-VEGF (vascular endothelial growth factor), bevacizumab; BRAFi; BRAF inhibitors; ICI, immune checkpoint inhibitors; TAS-102, trifluridine plus tipiracil hydrochloride.

**Figure 2 cancers-17-01007-f002:**
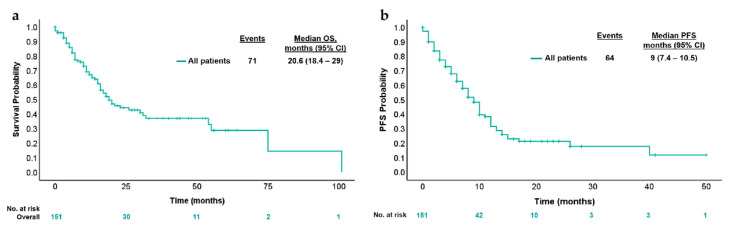
Kaplan–Meier estimates for (**a**) OS and (**b**) PFS according to first-line treatment. (**a**) Kaplan–Meier analysis of overall survival; (**b**) Kaplan–Meier analysis of progression-free survival.

**Figure 3 cancers-17-01007-f003:**
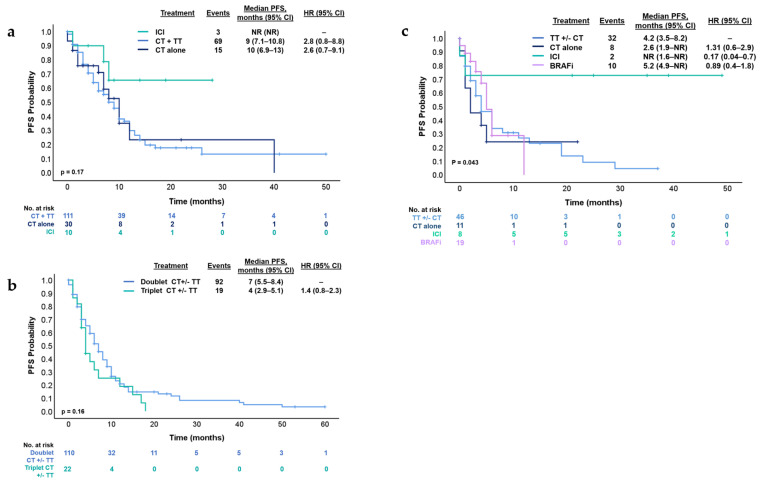
Kaplan–Meier estimates for (**a**) PFS according to treatment types in the first-line treatment. (**b**) PFS according to doublet chemotherapy or triplet chemotherapy in the first-line treatment. (**c**) PFS according to treatment type in the second-line treatment. BRAFi; BRAF inhibitors, CI, confidence interval; CT, chemotherapy; CT +/−, chemotherapy with or without; Doublet CT, XELOX, FOLFOX or FOLFIRI; HR, hazard ratio; ICI, immune checkpoint inhibitors; PFS, progression-free survival; Triplet CT, FOLFIRINOX; TT, targeted therapy (anti-EGFR or anti-VEGF).

**Figure 4 cancers-17-01007-f004:**
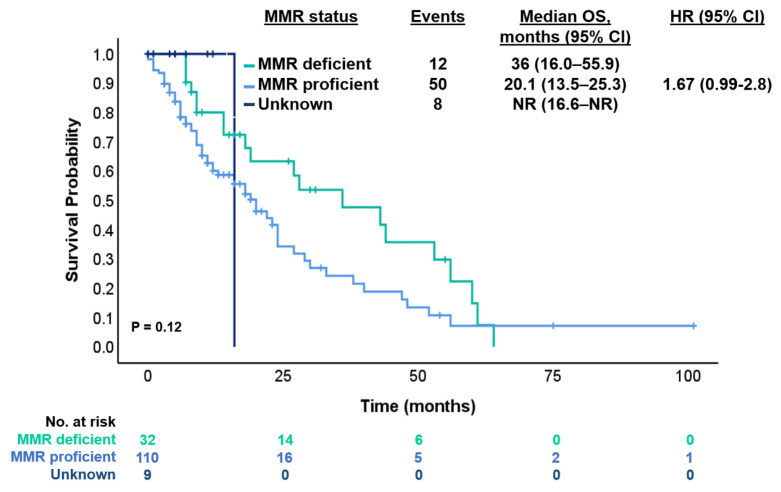
Kaplan–Meier estimates for OS according to status of MMR.

**Table 1 cancers-17-01007-t001:** Baseline site and patients’ characteristics.

Variable	Total = 161 n	(%)
Median age, years	58.3 (47–69)	
Sex		
Female	96	59.6%
Male	65	40.4%
ECOG *		
0–1	135	85.7
≥2	23	14.3
Tumor location		
Right	114	70.8%
Left	33	20.5%
Rectum	14	8.7
Stage at diagnosis		
I	1	0.6%
II	12	7.5%
III	33	20.5%
IV	115	71.4%
Histology		
Adenocarcinoma	112	69.5
Mucinous	43	26.8
Unknown	6	3.7
Number of metastasis site		
One	69	42.8
Two or more	78	48.5
Unknown	14	8.7
Site of metastasis		
Liver	90	55.9%
Nodes	82	50.9%
Lung	19	11.8%
Peritoneum	37	23%
Type of testing		
Primary tumor	144	89.5
Metastasis	8	4.9
Liquid biopsy	9	5.6
Status MMR **		
MMR-deficient	35	21.7%
MMR-proficient	117	72.7%
Unknown	9	5.6%
Status *RAS*		
Mutated	3	1.2%
Wild-type	134	83.8%
Unknown	24	15%
Status *BRAF*		
Before election 1st line	56	34.7%
Before election 2nd line	80	49.7%
After election 2nd or more line	25	15.6%
Surgery		
Curative	77	47.8%
Palliative	22	13.6%
No	62	72%
Lines of treatment received		
First	151	93.8%
Second	86	53.4%
Third	25	15.5%
Forth	10	6.2%
Fifth	3	1.8%
Sixth	2	1.2%

* ECOG, Eastern Cooperative Oncology Group; ** MMR, mismatch repair proteins.

**Table 2 cancers-17-01007-t002:** Factors predictive of progression-free survival in patients receiving first-line mCRC treatment (n = 151): Univariate and multivariate analyses.

VARIABLE	Progression or Death		Univariate Analysis		Multivariate Analysis	
	Yes n = 87 (%)	No n = 64 (%)	HR (95% CI)	*p*-Value	HR (95% CI)	*p*-Value
Age at diagnosis						
≤50 years	27 (62.8)	16 (37.2)	-		-	
50–70 years	44 (56.4)	34 (43–6)	0.71 (0.44–1.16)	0.18	0.76 (0.45–1.27)	0.29
>70 years	16 (53.4)	14 (46.6)	0.58 (0.31–1.08)	0.08	0.74 (0.37–1.43)	0.36
Gender						
Female	49 (55)	40 (45)	-		-	
Male	38 (61.3)	24 (38.7)	1.57 (1.01–2.4)	0.04	1.55 (0.98–2.45)	0.06
Stage at diagnosis					/	/
I	1 (100)	-	-			
II	7 (58.3)	5 (41.7)	0.15 (0.01–0.99)	0.08		
III	18 (54.5)	15 (45.5)	0.12 (0.01–0.72)	0.04		
IV	61 (58.1)	44 (41.9)	0.14 (0.01–0.86)	0.05		
Tumor location						
Left	61 (57.5)	45 (42.5)	-		-	
Right	14 (45.2)	17 (54.8)	1.12 (0.62–2.01)	0.71	0.96 (0.53–1.77)	0.09
Rectum	12 (85.7)	2 (14.3)	2.1 (1.12–3.9)	0.01	1.67 (0.37–3.3)	0.14
Histology					/	/
Adenocarcinoma	60 (54.6)	50 (45.4)	-			
Mucinous	23 (65.7)	12 (34.3)	0.76 (0.26–2.22)	0.61		
Diagnosis of the metastatic setting					/	/
Synchronous	66 (62.3)	40 (37.7)	0.94 (0.13–8.87)	0.96		
Metachronous	21 (46.7)	24 (53.3)	0.65 (0.09–4.84)	0.67		
Liver metastasis					/	/
No	34 (50.7)	33 (49.3)	-	-		
Yes	53 (63.1)	31 (36.9)	1.18 (0.77–2.72)	0.45		
Peritoneum metastasis						
No	65 (55.1)	53 (44.9)	-	-		
Yes	20 (64.5)	11 (35.5)	1.64 (0.99–2.72)	0.05	1.92 (1.12–3.28)	0.01
MMR status					/	/
pMMR	72 (60.5)	47 (39.5)	-	-		
dMMR	15 (46.9)	17 (53.1)	0.65 (0.38–1.16)	0.15		
Surgery primary tumor					1.79 (1.11–2.88)	0.01
No	72 (60.5)	46 (45.5)	1.8 (1.15–2.8)	0.009		
Yes	15 (46.9)	18 (36)	-			

## Data Availability

Individual clinical data and sequencing data cannot be shared publicly because the data consist of sensitive patient data. Data access are available upon request for researchers meeting confidentiality criteria through the Asociación Argentina de Oncología Clínica Ethics Committee, where the study was approved. All other data generated or analyzed during this study are included in the published article or can be obtained from the corresponding author upon reasonable request.

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
