# Peer review of "Patients with Colorectal Cancer and BRAFV600E-Mutation in Argentina: A Real-World Study—The EMOGI-CRC01 Study"

_cancers, 2025, doi:10.3390/cancers17061007_

Round 1

Reviewer 1 Report

Comments and Suggestions for Authors

The authors report the outcomes of BRAF mutation-positive colorectal cancer in Argentina. This study is interesting because it reflects the real world, with a variety of treatments due to the historical background and the high medical cost of BRAF inhibitors. The authors have several issues that need to be resolved before publication.

Major concerns

1. The prognosis of BRAF mutation-positive colorectal cancer is very poor. Comparing the triplet regimen with the doublet regimen, the survival curve tends to show a worse prognosis with PFS for the more potent regimen, the triplet regimen. What are the possible causes of this? The authors mention patient background (e.g., with or without dMMR) in their discussion, but was there a difference in patient background between the two groups?

2. Regarding PFS for second-line treatment, I think it reflects the real world that many cases in this study were not treated with BFAF inhibitors in terms of the historical background and medical costs. However, I feel that in this study, the contribution of BFAF inhibitors in prolonging PFS in the patients who used BFAF inhibitors is poor. With these results, I feel that the benefits of BRAF inhibitors are lacking.

 3. As for patient background in this study, Table 1 evaluates 161 patients. Of those, 151 patients were treated with first-line therapy. Are the cases that were not treated with first-line therapy Stage I or II cases? Or did the authors also provide first-line treatment for Stage I or II cases?

Author Response

Comments 1: The prognosis of BRAF mutation-positive colorectal cancer is very poor. Comparing the triplet regimen with the doublet regimen, the survival curve tends to show a worse prognosis with PFS for the more potent regimen, the triplet regimen. What are the possible causes of this? The authors mention patient background (e.g., with or without dMMR) in their discussion, but was there a difference in patient background between the two groups?

Response 1: [A total of 22 patients received the triplet regimen, compared to 110 patients who received the doublet regimen. Given the small sample size of the triplet group, statistical comparisons between the two regimens are limited in progression-free survival (PFS). Additionally, patients who received the triplet regimen had a higher tumor burden and more metastatic sites, which could have influenced their prognosis.

Comments 2: Regarding PFS for second-line treatment, I think it reflects the real world that many cases in this study were not treated with BFAF inhibitors in terms of the historical background and medical costs. However, I feel that in this study, the contribution of BFAF inhibitors in prolonging PFS in the patients who used BFAF inhibitors is poor. With these results, I feel that the benefits of BRAF inhibitors are lacking.

Response 2: Although the difference was not statistically significant, possibly due to the short follow-up time and the low number of patients who received BRAF inhibitors, the survival curves show a trend toward benefit in those who received BRAF inhibitors compared to those who did not. Additionally, the low proportion of patients who received BRAF inhibitors highlights the significant heterogeneity in treatment availability."

Comments 3: As for patient background in this study, Table 1 evaluates 161 patients. Of those, 151 patients were treated with first-line therapy. Are the cases that were not treated with first-line therapy Stage I or II cases? Or did the authors also provide first-line treatment for Stage I or II cases?

Response 3: The 10 patients were unable to start treatment due to poor performance status, requiring hospitalization, or because their oncologist considered them unfit for systemic therapy. We add this in **Treatment sequences**

Reviewer 2 Report

Comments and Suggestions for Authors

Greta Catani et al. reported a real-world study (EMOGI0CRC01) on CRC patients with BRAF V600E mutations. This analysis, derived from Latin America, provides critical real-world insights into the use of BRAFi within the local healthcare context. I have the following questions:

Here are my major concerns: 

1. Please explain why stage I–III (potentially curable patients) would require BRAF-targeted therapy. Did these patients undergo curative or palliative surgery? (Table 2 mentions surgery, but Table 1 does not.) Additionally, information on adjuvant therapy is crucial but missing in the report.

2. Analyzing survival outcomes (PFS and OS) for all stages together seems inappropriate. A stage-specific survival analysis is recommended. Moreover, the starting point for survival analysis is indicated as first-line therapy (lines 97–100, pages 2–3), but BRAFi is not always used as a first-line treatment. It would make more sense to calculate survival time from the initiation of BRAFi use no matter of the lines of BRAFi. Otherwise, the reported PFS and OS have no clear association with BRAFi usage, rendering subsequent analyses meaningless.

3. Details about immunotherapy monotherapy or combination therapy are lacking. Were all these cases dMMR? Please explain why the longest survival times were observed in this group per the regimens they received or the molecular patterns they have.

4. In the first paragraph of the Discussion, the authors mention two types of BRAF but fail to clearly explain BRAF mutation subtypes. It is unclear how this relates to the study's findings.

5. Lines 269–270 (page 9) state: "Notably, the median OS and PFS were higher in our cohort (20.6 and 9 months, respectively) in contrast to findings from previous reports." The authors do not provide reasons for this phenomenon. Please add a detailed explanation and expand on this in the Discussion section.

6. In the Discussion, the authors frequently propose that the survival differences in this report compared with previous studies are owing to the status of dMMR. However, when analyzing survival outcomes or conducting multivariate analyses, no significant statistical differences were observed. This weakens their argument. Please provide stronger and more convincing explanations in the section of Discussion.

7. In my opinion, discussing multiple mechanisms of resistance to BRAFi seems unrelated to the data reported in this study. The authors should consider whether this topic is worth including. Instead, it may be more valuable to discuss topics such as the clinical outcomes and treatment responses of dMMR groups, dMMR+BRAFi combinations, and RAS(mut)+BRAF(mut) groups. These are likely more relevant in a real-world study than unrelated resistance mechanisms.

8. The approval number of the IRBs is missing. Also, as the authors declared, this is an observational study, did you really get the written ICFs from the subjects? Please confirm.

Then, here are my minor concerns for this report, 

(1) Ensure consistency in numerical formatting throughout the manuscript, e.g., "20.6 months (line 29, page 1)" vs. "only 20% (line 30, page 1)." Use either one decimal place or no decimal places for all numbers, especially in Table 1. Additionally, unify the usage of percentages (%).

(2) In Figure 3c, "iBRAF" should be "BRAFi." Please confirm.

(3) In Table 1, the number of MMR-proficient cases is listed as "711." Please confirm.

(4) The percentage of cases with an unknown stage is 8.9%. Please verify this. The high proportion of missing data raises concerns. Explain how this was addressed to avoid introducing bias into the results.

(5) The higher prevalence of left-sided tumors requires a thorough explanation in the Discussion section.

Lines 204–205 indicate, "TT = TT, targeted therapy (anti-EGFR and anti-VEGF)." Confirm whether this refers to "and" or "or."

(6) The timing of BRAF testing is important—was it conducted before systemic therapy or after failure of frontline therapy?

(7) In line 268 (page 9), "BRAFV600E" should be italicized, with V600E as a superscript.

(8) In lines 290–291 (page 9), "Our results demonstrate better PFS with doublet chemotherapy than triplet (7 versus 4 months, respectively)." The p-value is missing.

Author Response

Thank you very much for taking the time to review and provide feedback on this manuscript. I greatly appreciate your insightful comments and suggestions. Please find below the detailed responses to your points, along with the corresponding revisions and corrections, which are highlighted in track changes in the re-submitted files.

Comments 1:  Please explain why stage I–III (potentially curable patients) would require BRAF-targeted therapy. Did these patients undergo curative or palliative surgery? (Table 2 mentions surgery, but Table 1 does not.) Additionally, information on adjuvant therapy is crucial but missing in the report.

Response 1: Nowadays, target therapy in earlier-stage is in Curative surgery was undergone in 47.8% (n=77) of the total patients included (n=161). Correction in table 1 (add information about surgery) and 3th paragraph in the result, we add information about adjuvant treatment. In the second paragraph of the discussion, we add information about target therapy trial in the earlier-setting.

**In our cohort, 28% of the included patients were diagnosed at an early-stage (stage I-III) but experienced metastatic progression during the course of the disease. For this reason, several ongoing clinical trials are investigating the use of BRAF-targeted therapies in this setting. These studies aim to determine whether incorporating targeted treatments can improve outcomes in this potentially curable population. The Alliance A022004 trial (NCT05710406), a phase II/III study evaluates whether the combination of encorafenib and cetuximab after surgery and chemotherapy; AIO-KRK-0420 (NCT05510895), a phase II trial investigates the efficacy of a neoadjuvant regimen combining encorafenib, binimetinib and cetuximab; and the NEORAF study (NCT05706779), a phase II evaluates the effectiveness of neoadjuvant treatment with encorafenib plus cetuximab in patients with operable BRAFV600E-mutant adenocarcinoma of the colon or upper rectum. The re-sults of these three trials are not publish yet but reflect a growing interest in integrating BRAF-targeted therapies into the treatment paradigm for earlier-stage BRAF-mutant CRC.**

Comments 2: Analyzing survival outcomes (PFS and OS) for all stages together seems inappropriate. A stage-specific survival analysis is recommended. Moreover, the starting point for survival analysis is indicated as first-line therapy (lines 97–100, pages 2–3), but BRAFi is not always used as a first-line treatment. It would make more sense to calculate survival time from the initiation of BRAFi use no matter of the lines of BRAFi. Otherwise, the reported PFS and OS have no clear association with BRAFi usage, rendering subsequent analyses meaningless.

Response 2: If it seems appropriate, I can remove the overall survival (OS) and only put on the manuscript progression-free survival (PFS) analysis for the entire population.

All patients included in the study are metastatic (either de novo or as part of disease progression), which may limit the generalizability of the results across all stages. I am open to revising the analysis to better align with the patient population and treatment lines. We agree that a more appropriate approach would be to calculate survival time from the initiation of BRAFi therapy, regardless of the treatment line. Due to the limited number of patients who received BRAF inhibitors (BRAFi), performing a stage-specific survival analysis would have been challenging. Given the small sample size, it is difficult to draw robust conclusions based solely on the use of BRAFi in different treatment lines.

Comments 3: Details about immunotherapy monotherapy or combination therapy are lacking. Were all these cases dMMR? Please explain why the longest survival times were observed in this group per the regimens they received or the molecular patterns they have.

Response 3:  We describe better the explanation about the ICI treatment.

3.4 Immunotherapy and BRAF inhibitors

Considering the total number of patients included in the treatment analysis (n=151), 15.2% (n=23) received ICI: 43.5% (n=10) in the first-line, 30.4% (n=7) in the second-line, and 26.1% (n=6) in the third-line setting. Among them, 93.1% (n=21) patients received pembrolizumab, while only 6.9% (n=2) received nivolumab/ipilimumab.

In the context of BRAF-mutant colorectal cancer, the activation of the MAPK signaling pathway, induced by the BRAF mutation, may contribute to increased expression of immune evasion-related proteins, such as PD-L1, thereby enhancing the tumor’s vulnerability to immunotherapy. Thus, the combination of a higher mutational burden in dMMR tumors and alterations in the MAPK signaling pathway facilitates the efficacy of immunotherapeutic agents, which, by interfering with immune checkpoints, allow for a more robust and effective immune response against the tumor.

Comments 4: In the first paragraph of the Discussion, the authors mention two types of BRAF but fail to clearly explain BRAF mutation subtypes. It is unclear how this relates to the study's findings.

Response 4: Thank you for the correction. We have modified this paragraph.

BRAFV600E-mutation mCRC is a highly aggressive and biologically heterogeneous disease, associated with poor prognosis and limited therapeutic options. In this re-al-world multicenter study, we analyzed the clinical characteristics, treatment strate-gies, and outcomes of 161 patients with BRAFV600E-mutation mCRC in Argentina. Our findings highlight significant variability in treatment approaches, with limited access to BRAF-targeted therapies in second- and third-line settings. Despite a mOS 20.6 months, disease progression remained a major challenge. Notably, while 93.8% of patients re-ceived first-line treatment, only 15.5% were able to undergo third-line therapy, reflect-ing the aggressive nature of the disease and the rapid clinical deterioration that limits the feasibility of subsequent treatment lines.

Comments 5: Lines 269–270 (page 9) state: "Notably, the median OS and PFS were higher in our cohort (20.6 and 9 months, respectively) in contrast to findings from previous reports." The authors do not provide reasons for this phenomenon. Please add a detailed explanation and expand on this in the Discussion section.

Comments 6: In the Discussion, the authors frequently propose that the survival differences in this report compared with previous studies are owing to the status of dMMR. However, when analyzing survival outcomes or conducting multivariate analyses, no significant statistical differences were observed. This weakens their argument. Please provide stronger and more convincing explanations in the section of Discussion.

Response 5-6: The higher median overall survival (OS) and progression-free survival (PFS) observed in our cohort compared to previous reports may be attributed to several statistical and methodological factors. First, our cohort may have included a more favorable patient population, such as those with better baseline performance status or more localized disease at diagnosis, which could have led to better outcomes. Additionally, the size of our cohort, while sufficient for meaningful analysis, may have resulted in a more homogeneous group in terms of treatment adherence and follow-up, which can influence survival outcomes. Also, the majority of the patients included in our cohort were followed up in the private healthcare sector, where patient monitoring tends to be more stringent. Unfortunately, in our country, there is a significant gap between the public and private healthcare systems, particularly regarding the availability of human resources and physical resources, such as medications, hospital infrastructure, medical equipment, and facilities, which can directly impact patient care and survival outcomes.

Comments 7: In the Discussion, the authors frequently propose that the survival differences in this report compared with previous studies are owing to the status of dMMR. However, when analyzing survival outcomes or conducting multivariate analyses, no significant statistical differences were observed. This weakens their argument. Please provide stronger and more convincing explanations in the section of Discussion.

Response 7: Thank you for your valuable comment. I agree with your observation regarding the survival differences and the role of dMMR status. We acknowledge that the lack of statistically significant differences in the survival analysis, is likely due to the small follow up of dMMR patients in our cohort, which limits the statistical power to detect a significant effect.

Comments 8: The approval number of the IRBs is missing. Also, as the authors declared, this is an observational study, did you really get the written ICFs from the subjects? Please confirm.

Response 8: Although this is a retrospective study, we confirm that all patients and/or their families signed informed consent forms (ICFs) to participate in the study and to allow the use of their data. This study involved the participation of several centers across Argentina and began several years ago. During the initial phase of the project, informed consent was collected from all participants, ensuring that their participation was voluntary and in accordance with ethical guidelines.  

Round 2

Reviewer 1 Report

Comments and Suggestions for Authors

The presented manuscript is revised adequately. There is one point that needs to be confirmed. I understand the response to comment 2: In the BRAF inhibitor survival curves (Figure 2,c), the curve for the BRAF inhibitor is represented in purple. However, the BRAF inhibitor in the No. at risk tier is green. Please check this section.

Author Response

Thank you very much for your time and effort invested in our manuscript. In this final version, I have reinstated the OS analysis results along with the corresponding figure and changed the colours of the Graphic that you suggest. 
